# LIVE: Learnable In-Context Vector for Visual Question Answering

**Yingzhe Peng**[1,2]**, Chenduo Hao**[1,2]**, Xinting Hu**[3]**, Jiawei Peng**[1, 2]**, Xin Geng**[1,2]**, Xu Yang**[1,2*]

[1] Southeast University
[2] Key Laboratory of New Generation Artificial Intelligence Technology and
Its Interdisciplinary Applications (Southeast University), Ministry of Education, China
`{yingzhe.peng, 213201447, pengjiawei, xgeng, xuyang_palm}@seu.edu.cn`
[3] Nanyang Technological University
`xinting001@e.ntu.edu.sg`

## Abstract

As language models continue to scale, Large Language Models (LLMs) have exhibited emerging capabilities in In-Context Learning (ICL), enabling them to solve language tasks by prefixing a few in-context demonstrations (ICDs) as context. Inspired by these advancements, researchers have extended these techniques to develop Large Multimodal Models (LMMs) with ICL capabilities. However, applying ICL usually faces two major challenges: 1) using more ICDs will largely increase the inference time and 2) the performance is sensitive to the selection of ICDs. These challenges are further exacerbated in LMMs due to the integration of multiple data types and the combinational complexity of multimodal ICDs. Recently, to address these challenges, some NLP studies introduce non-learnable In-Context Vectors (ICVs) which extract useful task information from ICDs into a single vector and then insert it into the LLM to help solve the corresponding task. However, although useful in simple NLP tasks, these non-learnable methods fail to handle complex multimodal tasks like Visual Question Answering (VQA). In this study, we propose Learnable In-Context Vector (LIVE) to distill essential task information from demonstrations, improving ICL performance in LMMs. Experiments show that LIVE can significantly reduce computational costs while enhancing accuracy in VQA tasks compared to traditional ICL and other non-learnable ICV methods. The code is available at `https://github.com/ForJadeForest/LIVE-Learnable-In-Context-Vector`.

## 1 Introduction

As language models continue to scale up, Large Language Models (LLMs) [1–3] have demonstrated emerging capabilities in In-Context Learning (ICL) [4]: these models can solve language tasks when provided with a few similar examples, termed in-context demonstrations (ICDs), as context. Unlike traditional task-specific fine-tuning, ICL, a efficient method to adapt LLM to downstream task [5, 6], achieves comparable performance without necessitating updates to millions or trillions of model parameters [7]. By prefixing just a handful of data samples to the query input, ICL configures a model's behavior to produce the corresponding output, thus facilitating rapid adaptation across a wide range of downstream tasks. Inspired by these advancements in the language domain, researchers have extended these techniques to develop Large Multimodal Models (LMMs) with ICL capabilities [8–10].

---

[1]Corresponding author.

38th Conference on Neural Information Processing Systems (NeurIPS 2024).

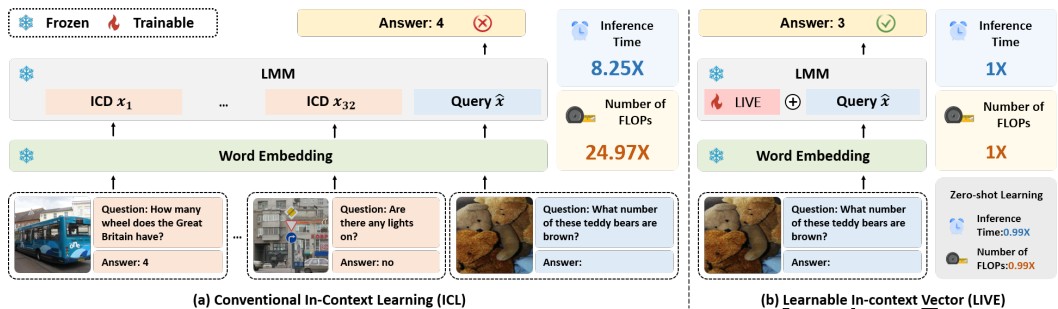

Figure 1: (a) Conventional ICL is more sensitive to the ICD selection and requires more inference time. (b) LIVE is more robust and reduces inference time by inputting a shift vector.

Employing ICL in LLMs meets two challenges: Firstly, although increasing the number of ICDs typically enhances performance [7], this practice conflicts with computational efficiency constraints. As ICDs are prefixed to the query, the increase in input tokens severely impacts the Transformer's inference speed, causing a marked slowdown in computational performance. Secondly, the effectiveness of ICL is vulnerable to the selection of demonstrations [4, 11–13], particularly when only a limited number are used. It makes the process of choosing demonstrations critical for optimal performance. However, developing selection strategies and measuring the effectiveness of ICDs remain open questions [14–17]. For LMMs, the challenges above are further exacerbated: 1) The computational complexity is significantly increased due to the integration of multiple data types as ICDs (as shown in Figure 1(a)). 2) The task of selecting effective multi-modal ICDs becomes more complex and nuanced, as each modality contributes uniquely to understanding the context [18, 19], further complicating the assessment of their combined effect in demonstration selection.

To alleviate these two challenges, recent research on LLMs has introduced In-Context Vector (ICV) to extract the most useful task information from ICDs, and then use it to directly influence the processing in LLMs [20–22]. For example, [20] proposes that by using multiple demonstrations and a dummy query as inputs, the representation of the last token from a middle layer of the model can be extracted as the vector. This vector is then used to replace the representation of the corresponding token in the same layer during inference, which can achieve performance comparable to ICL. Such in-context vector alleviates the requirement of multiple ICDs during inference, as well as effectively bypasses the complexity of the individual selection of demonstrations by representing the most effective components across many demonstrations.

However, these studies apply non-learnable strategies to extract ICVs, although useful in some simple NLP tasks, lose the efficacy in complex multi-modal tasks like Visual Question Answering (VQA). Our preliminary experiments have demonstrated that directly applying these non-learnable ICVs yields unsatisfactory results. The principal reason is the intrinsic complexity of VQA compared to the language tasks addressed by these non-learnable ICVs. For example, the previous methods focus on simple NLP task, such as Antonym [23] and Country-Capital [21], whose distribution patterns can be easily identified by LLMs. In contrast, as a unified vision-language task, VQA encompasses a diverse array of question types, where each one corresponds to a different vision-understanding task. For instance, questions like "What is this?" or "How many are there?" require classification and counting abilities, respectively. These varied requirements imply that the task information, which non-learnable methods attempt to abstract, cannot be effectively captured by a single ICV.

In this study, to make ICVs abstract more useful VQA task information, we try to distill the task information implied in demonstrations into a single **Learnable In-Context Vector** (LIVE). Our method is motivated by the observation [20] that ICL can be treated as a process of "shifting" the direction of the latent states of query towards the target, *i.e.*, adding this latent state with a shift vector. Then we hope to learn suitable ICVs to replace the ICDs during inference to shift the direction. To achieve this, we train LIVE by minimizing the output distributions of a LMM got by only using LIVE and by inputting a few demonstrations. During training, we use different 32-shot randomly sampled demonstrations for different queries to distill task knowledge. Then LIVE is encouraged to capture the most essential task information from these different combinations by removing the individual characteristics of demonstrations. Moreover, [24] finds that during ICL, different layers of LLM

have diverse roles in addressing demonstrations. Then in our method, each layer is assigned with a unique ICV to capture more fine-grained task information.

Our LIVE inherits the efficiency of previous non-learnable ICVs, *i.e.*, during inference, under the same performance conditions, LIVE only needs 1/24.97 FLOPs number of 32-shot ICL. Additionally, in VQAV2/OKVQA, LIVE improves accuracy by 2.36/1.6 compared to 32-shot ICL. We also compare LIVE to LoRA [25] that when comparable trainable parameters are used, LIVE requires much fewer training samples than LoRA (500 vs. 8000) to achieve satisfactory performance. Besides, we design lots of analytical experiments to validate whether LIVE can better shift the hidden states of queries to the target direction and analyze why previous non-learnable methods fail to solve VQA.

## 2 Related Work

**In-Context Vector:** Recently, more and more researchers in NLP have begun to focus on using an In-Context Vector (ICV) to modify the activation values during the forward propagation of LLM to simulate the effect of ICDs in ICL. [20] propose the "Task Vector", which extracts the representation of the middle layer from the LLM during ICL inference as the ICV, and replaces the representation of the same layer during zero-shot inference. Meanwhile, [21] introduced the "Function Vector", which uses attention weight analysis to take the mean of the activation values of the attention heads that most significantly affect the final result in ICL inference as the final ICV. This vector is then directly added to the representation of the middle layer during zero-shot inference to form a new representation. On the other hand, [22] propose "PCA In-Context Vector". They believe that the ICV should be closer to the LLM's representation of the task output and farther from the task input representation. Thus, they extract the input and output representations of several demonstrations and using PCA to find the overall principal direction as the ICV. These efforts mainly focus on using non-learnable methods to find the specific ICV for NLP tasks, achieving effects similar to ICL in various tasks. However, these methods only are tested on some simple tasks in NLP. When LMMs face with more complex tasks, the performance of these methods remains uncertain.

**ICL in LLM:** Prompt engineering allows LLMs to tackle specific tasks without requiring fine-tuning [26–33]. A specific form of this approach, ICL, further improves these capabilities by creating prompts that include several demonstrations. ICL has already demonstrated superior performance and good generalization on many tasks [34, 7, 35, 36], and can be easily adapted to downstream tasks. However, the use of ICL faces several issues: first, ICL is very sensitive to the selection and arrangement order of demonstrations [4, 11–13, 37–39]; poor demonstrations can severely impact ICL performance. Second, too many demonstrations can significantly slow down the inference speed of LLMs [40]. While ICVs can effectively address these two issues, as it can use only queries as input to the model while preserving ICL performance, without the need for demonstrations as input.

**ICL in LMM:** As the performance of LLMs continues to improve, an increasing number of researchers begin to adapt LLMs to the multimodal domain [41–46]. Relying on the powerful inference capabilities of LLMs, some LMMs have started to exhibit ICL capabilities, such as Flamingo [8] and IDEFICS [9]. Moreover, these models have significantly enhanced their ICL capabilities by concatenating multiple samples as contextual information during the training process. Currently, researchers mainly focus on how to configure demonstrations to address the sensitivity of ICL performance in LMMs. [18, 19] have respectively adopted heuristic retrieval methods for selecting demonstrations in Image Captioning and VQA. However, no researchers have yet extracted ICV from LMMs and evalutaed it. Therefore, the effectiveness of ICV in LMMs still needs further exploration. Considering that the IDEFICS model shares the same model structure as Flamingo and possesses stronger ICL capabilities, we primarily focus on valid our method on the IDEFICS model.

## 3 LIVE: Learnable In-Context Vector

Here we show how to derive the formulation of the shift vector from Self-Attention (SA) mechanism and then introduce how to design LIVE based on this formulation. Generally, to implement In-Context Learning (ICL) using a language or multimodal model (LLM/LMM) $\mathcal{M}$, the input has the following form: $\boldsymbol{X} = \{\boldsymbol{X}_D, \hat{\boldsymbol{x}}\}$, where $\boldsymbol{X}_D = \{\boldsymbol{x}_1, ..., \boldsymbol{x}_k\}$ represents the concatenation of $k$ In-Context Demonstrations (ICDs), and $\hat{\boldsymbol{x}}$ denotes the query input, as shown in Figure 2. Given $\boldsymbol{X}$ as *Key* and

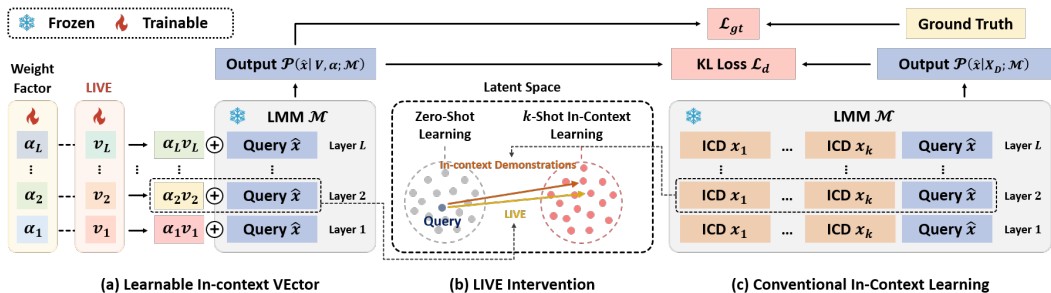

Figure 2: The LIVE training pipeline: (a) The distribution $\mathcal{P}(\hat{\boldsymbol{x}}|\boldsymbol{V}, \boldsymbol{\alpha}; \mathcal{M})$ of LMMs output when using LIVE. (b) Adding LIVE into the representations of the query to simulate the shift effect brought by demonstrations. (c) The distribution $\mathcal{P}(\hat{\boldsymbol{x}}|\boldsymbol{X}_D; \mathcal{M})$ of LMMs output when using demonstrations.

*Value*, for each token $\hat{x}_i$ of $\hat{\boldsymbol{x}}$, applying *Self-Attention (SA)* once yields:

$$\mathrm{SA}(\hat{x}_i, \boldsymbol{X}, \boldsymbol{X}) = \mathrm{SA}(\hat{x}_i, \begin{bmatrix} \boldsymbol{X}_D \\ \hat{\boldsymbol{x}} \end{bmatrix}, \begin{bmatrix} \boldsymbol{X}_D \\ \hat{\boldsymbol{x}} \end{bmatrix}) = \mathrm{softmax}([\hat{x}_i \boldsymbol{X}_D^\top \quad \hat{x}_i \hat{\boldsymbol{x}}^\top]) \begin{bmatrix} \boldsymbol{X}_D \\ \hat{\boldsymbol{x}} \end{bmatrix}, \quad (1)$$

where vector $[\hat{x}_i \boldsymbol{X}_D^\top \quad \hat{x}_i \hat{\boldsymbol{x}}^\top] \in \mathbb{R}^{1 \times l}$ and $l$ denotes the sequence length of the entire input $[\boldsymbol{X}_D, \hat{\boldsymbol{x}}]$. Expanding the softmax function, we obtain:

$$\mathrm{softmax}([\hat{x}_i \boldsymbol{X}_D^\top \quad \hat{x}_i \hat{\boldsymbol{x}}^\top]) = \left[ \underbrace{\frac{\exp(\hat{x}_i \boldsymbol{X}_D^\top)_1}{Z_1 + Z_2}, \ldots, \frac{\exp(\hat{x}_i \boldsymbol{X}_D^\top)_{l_c}}{Z_1 + Z_2}}_{\text{expansion of } \hat{x}_i \boldsymbol{X}_D^\top}, \underbrace{\frac{\exp(\hat{x}_i \hat{\boldsymbol{x}}^\top)_1}{Z_1 + Z_2}, \ldots, \frac{\exp(\hat{x}_i \hat{\boldsymbol{x}}^\top)_{l_q}}{Z_1 + Z_2}}_{\text{expansion of } \hat{x}_i \hat{\boldsymbol{x}}^\top} \right]$$

$$= \left[ \frac{\exp(\hat{x}_i \boldsymbol{X}_D^\top)}{Z_1 + Z_2} \quad \frac{\exp(\hat{x}_i \hat{\boldsymbol{x}}^\top)}{Z_1 + Z_2} \right], \quad (2)$$

where $l_c$ and $l_q$ represent the lengths of the $\boldsymbol{X}_D$ and $\hat{\boldsymbol{x}}$, respectively. $Z_1$ and $Z_2$ are the sum of exponential scores between the query token $\hat{x}_i$ with each token in $\boldsymbol{X}_D$ and $\hat{\boldsymbol{x}}$: $Z_1 = \sum_{l_c} \exp(\hat{x}_i \boldsymbol{X}_D^\top)$ and $Z_2 = \sum_{l_q} \exp(\hat{x}_i \hat{\boldsymbol{x}}^\top)$. This leads to the following formulation of SA:

$$\begin{aligned}
\mathrm{SA}(\hat{x}_i, \boldsymbol{X}, \boldsymbol{X}) &= \frac{\exp(\hat{x}_i \boldsymbol{X}_D^\top) \boldsymbol{X}_D}{Z_1 + Z_2} + \frac{\exp(\hat{x}_i \hat{\boldsymbol{x}}^\top) \hat{\boldsymbol{x}}}{Z_1 + Z_2} \\
&= \frac{Z_1}{Z_1 + Z_2} \frac{\exp(\hat{x}_i \boldsymbol{X}_D^\top) \boldsymbol{X}_D}{Z_1} + \frac{Z_2}{Z_1 + Z_2} \frac{\exp(\hat{x}_i \hat{\boldsymbol{x}}^\top)}{Z_2} \\
&= \frac{Z_1}{Z_1 + Z_2} \mathrm{softmax}(\hat{x}_i \boldsymbol{X}_D^\top) \boldsymbol{X}_D + \frac{Z_2}{Z_1 + Z_2} \mathrm{softmax}(\hat{x}_i \hat{\boldsymbol{x}}^\top) \hat{\boldsymbol{x}} \\
&= \mu \mathrm{SA}(\hat{x}_i, \boldsymbol{X}_D, \boldsymbol{X}_D) + (1 - \mu) \mathrm{SA}(\hat{x}_i, \hat{\boldsymbol{x}}, \hat{\boldsymbol{x}}),
\end{aligned} \quad (3)$$

where $\mu = Z_1/(Z_1 + Z_2)$. Let $h(z) = \mathrm{SA}(\hat{x}_i, z, z)$. The output of $\mathrm{SA}(\hat{x}_i)$ can then be expressed as:

$$\mathrm{SA}(\hat{x}_i, \boldsymbol{X}, \boldsymbol{X}) = \mu h(\boldsymbol{X}_D) + (1 - \mu) h(\hat{\boldsymbol{x}}) \quad (4)$$

As noted in Equation 4, we observe that $h(\hat{\boldsymbol{x}})$ is the representation obtained with self-attention over the query $\hat{\boldsymbol{x}}$ without appending any ICD; $h(\boldsymbol{X}_D)$ functions similarly to a "shift" vector, altering the attention representation $h(\hat{\boldsymbol{x}})$ by incorporating contextual information from the ICDs $\boldsymbol{X}_D$. The coefficient $\mu$ quantifies the degree of influence $X_D$ has over the original query representation. For a visual demonstration of how ICDs shift the representation space, see Figure 2 (b). Consequently, once learning a general shift direction to replace the effect of $h(\boldsymbol{X}_D)$, we can employ this shift direction to simulate the ICL process of LMMs without actual demonstrations.

We propose a novel method that involves a **Learnable In-Context Vector** (LIVE) to simulate the ICL process without actual demonstrations. This approach aims to abstract general task information from demonstrations, enabling it to shift the model's representation toward the direction influenced by the ICDs. Figure 2 shows the training pipeline of LIVE. The LIVE training dataset, denoted as $\mathcal{D} = \{\boldsymbol{d}_1, \ldots, \boldsymbol{d}_N\}$, is a subset of the VQA dataset training split, created by randomly selecting $N$ question-answer pairs from it. We use each training sample $\boldsymbol{d}_i$ to simulate the query sample $\hat{\boldsymbol{x}}$ in ICL, and randomly select $k$ demonstrations from $\mathcal{D} \setminus \{\boldsymbol{d}_i\}$ for it. Additionally, [24] shows that during

ICL, each layer of an LLM performs a distinct role. Motivated by this, we assume that for LMM, each layer also requires a specific shift direction. We assign a learnable vector $v_l$ and a weight factor $\alpha_l$ for each layer $l$ to learn the unique shift effect. Our final LIVE comprises of the vector set $V$ and the corresponding weight factor set $\alpha$ as:

$$V = \{v_1, v_2, ..., v_L\}, \quad v_i \in \mathbb{R}^{1 \times d}$$
$$\alpha = \{\alpha_1, \alpha_2..., \alpha_L\}, \quad \alpha_i \in \mathbb{R}^{1 \times 1}, \tag{5}$$

where $L$ is the number of layers. To train $V$ and $\alpha$, we align the distribution of the model's outputs for the query when shifted by demonstrations, $\mathcal{P}(\hat{x}|X_D; \mathcal{M})$, with that shifted by our LIVE, $\mathcal{P}(\hat{x}|V, \alpha; \mathcal{M})$. This alignment is achieved by minimizing the Kullback-Leibler (KL) divergence:

$$\mathcal{L}_d = \text{KL}(\mathcal{P}(\hat{x}|X_D; \mathcal{M}) \,||\, \mathcal{P}(\hat{x}|V, \alpha; \mathcal{M})) \tag{6}$$

To obtain the distribution $\mathcal{P}(\hat{x}|X_D; \mathcal{M})$, for each query $\hat{x}$, we randomly select $k$ demonstrations to form $X_D$. These are concatenated with the query to form the inputs for the model. The model's output for the query is then considered as the shifted distribution $\mathcal{P}(\hat{x}|X_D; \mathcal{M})$.

To obtain the output of $\hat{x}$ by using LIVE, we follow [20, 22], we use the vector $v_l$ to shift the each layer's output representation $\mathcal{M}_l(\hat{x}_i)$ and get: $\mathcal{M}_l(\hat{x}_i)' = \mathcal{M}_l(\hat{x}_i) + \alpha_l v_l$, which is shown in Figure 2(a). After applying LIVE to shift the representations at each layer, we obtain the output distribution $\mathcal{P}(\hat{x}|V, \alpha; \mathcal{M})$. Notably, during training, $X_D$ for each query $\hat{x}$ includes randomly sampled 32-shot demonstrations. This strategy encourages our LIVE to extract the most useful common information from various demonstration combinations and prevents it from being influenced by the individual characteristics of certain demonstrations.

In addition, to facilitate the LIVE in acquiring more task-specific information, we also optimize the $\mathcal{P}(\hat{x}|V, \alpha; \mathcal{M})$ with the ground truth by $\mathcal{L}_{\text{gt}}$. Thus, the overall loss $\mathcal{L}$ is defined as:

$$\mathcal{L} = \lambda \mathcal{L}_{\text{gt}} + \mathcal{L}_d \,,$$
$$\text{where } \mathcal{L}_{\text{gt}} = -\sum_i \log \mathcal{P}(\hat{x}_i \mid V, \alpha; \mathcal{M}) \,. \tag{7}$$

where $\lambda$ is the hyper-parameter to control the importance of ground truth loss.

## 4 Experiments

### 4.1 Setting and implementation details

**Model and Dataset:** We evaluate our approach using the IDEFICS-9B model [9] across two datasets: VQAv2 [47] and OKVQA [48]. **VQAv2** emphasizes open-ended VQA tasks, encompassing $4,437,570$ question-answer pairs in its training split, supplemented by an additional $2,143,540$ pairs in the validation split. **OKVQA** is a large-scale dataset designed for models that require external knowledge to answer questions. It consists of $14,055$ question-answer pairs, with $9,009$ allocated for training and $5,046$ for validation. For both VQAv2 and OKVQA datasets, We train our LIVE on $8,000$ pairs from each training set. Due to computational resource limitations, we randomly sample $10,000$ question-answer pairs from the VQAv2 validation split for evaluation [18]. For OKVQA, we utilize the entire validation split.

**LIVE Setting:** During training, we assign 32-shot demonstrations for each query, enabling LIVE to acquire better directions of shifting vectors for VQA tasks. The $v_i$ is initialized using a normal distribution with a mean of $0$ and a standard deviation of $0.01$, and all $\alpha_i$ are initialized to $0.1$. More detailed training parameters can be found in Appendix.

### 4.2 Results

#### 4.2.1 Compared Methods

We primarily compare the following methods:

**Zero-Shot**: The model uses only the query as input.

**k-Shot ICL**:The model uses $k$ demonstrations, randomly selected from the VQA dataset training split, along with the query as input.

Table 1: Accuracy (%) with Different ICVs Methods and Finetuning Methods, where numbers in parentheses indicate multiples of LIVE trainable parameters.

| | Zero-Shot | 32-shot ICL | TV | FV | PCA-ICV | LoRA | LIVE (Ours) |
|---|---|---|---|---|---|---|---|
| VQAv2 | 29.25 | 56.18 | 43.68 | 30.21 | 34.75 | 49.02 | **58.54** |
| OKVQA | 30.54 | 48.48 | 32.68 | 31.02 | 30.59 | 34.21 | **50.08** |
| Total Trainable Parameters | - | - | - | - | - | $1,155,136(\times 8.8)$ | $131,104(\times 1.0)$ |

**Non-Learnable ICV Methods**: We extend three established non-learnable ICV methods from language models to our multimodal settings: (1) **Task Vector (TV) [20]** uses $k$ demonstrations and a dummy query to extract the representation of the last token from a middle layer of the model as the ICV. During inference, this vector replaces the representation of the last token in the same layer. We conduct evaluations on **TV** by implementing it across various layers and select the layer where it achieves the highest performance improvement. (2) **Function Vector (FV) [21]** employs a small subset of the validation data to derive the mean output from critical attention heads, forming the ICV. During inference, this vector is added to the representations of the last token within a specific layer. We conduct evaluations on **FV** by implementing it across various layers and select the layer where it achieves the highest performance improvement. (3) **PCA In-Context Vector (PCA-ICV) [22]** computes the ICV by applying PCA to the difference between the question and question-answer representations from $k$ demonstrations. During inference, these vectors are added to the representations of all tokens at each layer,

**LoRA [25]**: This method finetunes the LMMs with the same number of samples of training LIVE. We add the LoRA module in the token classification head of the last layer. In this way, the number of trainable parameters is comparable to that of LIVE.

### 4.2.2 Performance and Inference Efficiency on VQA.

We present performance comparisons with various methods in Table 1. Certain existing methods show only marginal improvements over Zero-Shot baselines, *e.g.*, FV improves by 0.96/0.48 on VQAv2/OKVQA and PCA-ICV improves by 0.04 on OKVQA. Besides, we observe that all the previous non-learnable ICV methods do not reach the performance of the standard 32-shot ICL, *e.g.*, the best non-learnable method, TV, is still 12.5/15.8 lower than 32-shot ICL on VQAv2/OKVQA. In contrast, our LIVE achieves an accuracy improvement of 2.36 on VQAv2 and 1.6 on OKVQA over 32-shot ICL. These results highlight the inefficacy of non-learnable methods in capturing essential task-specific information for VQA, whereas LIVE, by leveraging diverse 32-shot ICL demonstrations for each query during training, manages to abstract useful task information effectively. We further show that our LIVE outperforms LoRA with less trainable parameters, suggesting LIVE can abstract task information more efficiently.

Figure 3 displays the efficiency of LIVE during inference compared to other methods. We average the FLOPs and actual inference time consumption per forward pass over 1000 randomly sampled queries.[1] We observe that LIVE only needs 1/24.97 FLOPs and 1/8.25 inference time of 32-shot ICL per forward pass. Additionally, LIVE maintains almost the same inference speed as Zero-Shot. These comparisons validate the efficiency of LIVE during inference.

### 4.3 Ablation Studies

We use ablation studies to explore the effects of diverse settings, including different training losses, the shot number of demonstrations $k$ used during training, and the number of training data $N$.

**Training Loss:** Table 2 compares the results of using different losses: only $\mathcal{L}_{gt}$ in Eq. (7) or $\mathcal{L}_d$ in Eq. (7). We find that only using $\mathcal{L}_{gt}$ (same as standard fine-tuning) significantly damages the performance, *e.g.*, $\mathcal{L}_{gt}$ achieves 16.9/6.12 lower accuracy on VQAv2/OKVQA compared to using the combined loss $\mathcal{L}$; yet using only $\mathcal{L}_d$ results in a smaller performance drop – 3.78/3.14 lower VQAv2/OKVQA than when using $\mathcal{L}$. This suggests that with a limited number of trainable parameters, LIVE trained with $\mathcal{L}_d$ is more robust and capable of capturing essential

---

[1]For detailed hardware information, refer to Appendix.

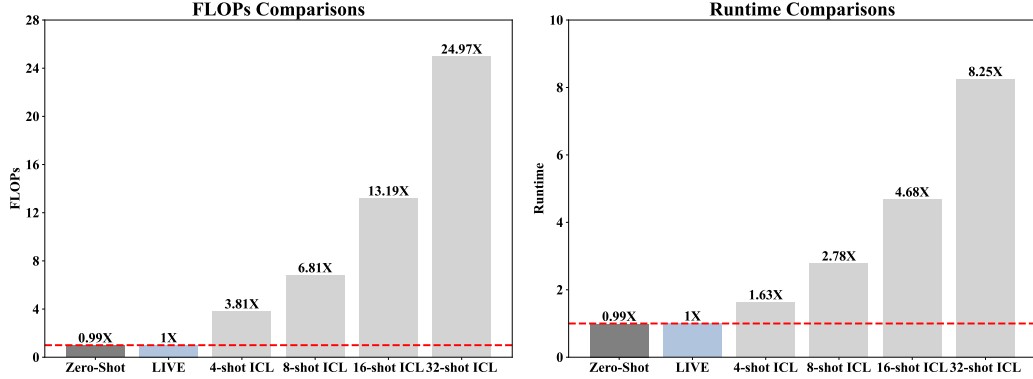

Figure 3: The total number of FLOPs and real inference time consumption of ICL, Zero-Shot, LIVE for 1000 query samples.

task information than $\mathcal{L}_{gt}$. It underscores that the LIVE cannot solely rely on fine-tuning with LMMs on specific datasets, but should effectively leverage abstracted insights from demonstrations.

**Number of Demonstrations** $k$: We compare the performance of LIVE trained with $k$ demonstrations per query and the corresponding $k$-shot ICL in Table 3. The result shows that an increase in the number of demonstrations enhances the performance of ICL and LIVE, indicating that more demonstrations can provide each query with a richer context to help train LIVE. Additionally, LIVE consistently surpasses the performance of $k$-shot ICL across different training sizes, showcasing the robustness of our LIVE in utilizing demonstrations. Notably, when the number of demonstrations is limited, the performance gap between LIVE and ICL becomes more pronounced. This is because ICL is highly sensitive to the choice of demonstrations; with insufficient demonstrations, the model may shift the query representations in an incorrect direction. In contrast, LIVE continuously by learning the main shift direction of the query representations from the demonstrations, reduces the negative impact of poor demonstrations on the query during training and is more robust that can extract essential task information.

Table 2: Accuracy (%) of LIVE with Different Training Loss on VQA.

|  | $\mathcal{L}_d$ | $\mathcal{L}_{gt}$ | $\mathcal{L}$ |
|---|---|---|---|
| VQAv2 | 54.76 | 41.64 | 58.54 |
| OKVQA | 46.94 | 43.96 | 50.08 |

**Size of Training Set:** Figure 4 illustrates how varying the number of training samples impacts the performance of LIVE and LoRA. On the VQAv2 dataset, both methods show improved performance with increasing data sizes. Notably, LIVE performs exceptionally well across both low and high training sizes. It achieves performance close to that of 1-shot ICL with just 700 training samples and surpasses 32-shot ICL with 4,000 training samples. In contrast, LoRA does not exceed the performance of 1-shot ICL, even when expanded to 8,000 samples. For OKVQA,

Table 3: Accuracy (%) of Different Number of Demonstrations on VQA.

| Task | Method | Number of Demonstrations | | | | |
|---|---|---|---|---|---|---|
|  |  | 1 | 4 | 8 | 16 | 32 |
| VQAv2 | LIVE | **56.84** | **57.60** | **58.25** | **58.27** | **58.54** |
|  | ICL | 51.39 | 53.72 | 54.24 | 55.70 | 56.18 |
| OKVQA | LIVE | **47.51** | **47.68** | **49.40** | **49.71** | **50.08** |
|  | ICL | 40.75 | 46.11 | 46.79 | 47.70 | 48.48 |

the performance of LoRA with small data sizes is even worse than Zero-Shot. This is because OKVQA requires external knowledge to answer questions, while learning external knowledge from a small amount of data can disrupt the inherent knowledge of the pre-trained model, leading to a significant drop in performance. Conversely, LIVE excels by focusing on learning shift direction, thus preserving the model's inherent reasoning abilities. With just 500 samples, LIVE outperforms 1-shot ICL, and with 4,000 samples, it nearly matches the performance of 32-shot ICL. These observations underscore LIVE's superior efficiency over LoRA in capturing and utilizing complex reasoning capabilities with much fewer training samples.

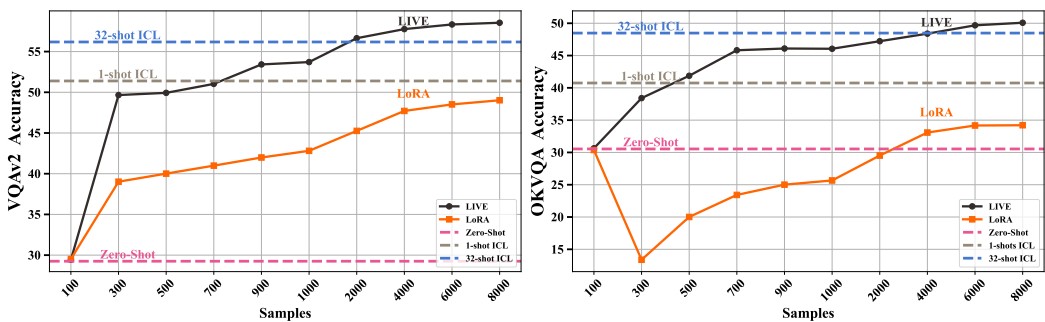

Figure 4: Accuracy (%) of LIVE and LoRA with different size of training set.

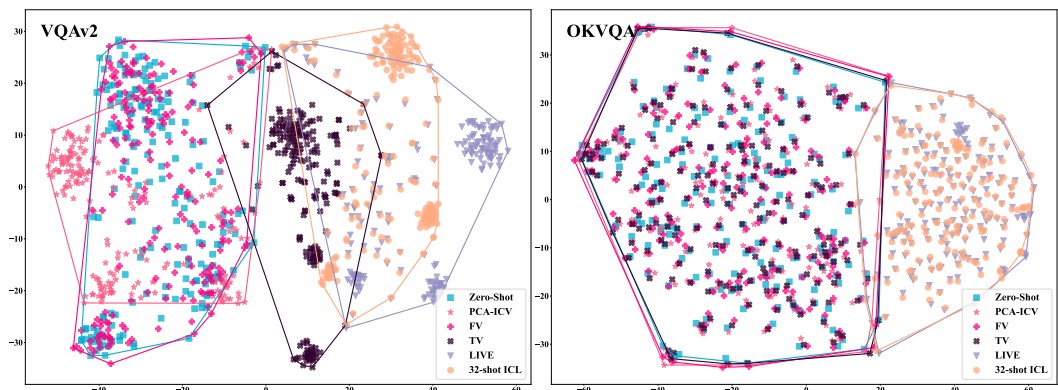

Figure 5: T-SNE visualization of first answer token representations over 200 queries.

## 4.4 Analysis

### 4.4.1 The Shifting Effect in Latent Space

To better demonstrate the shifting effect of LIVE on query samples, we randomly select 200 query samples and conduct different methods of inference in LMMs. We extract the representation vector of the first answer token for T-SNE dimensionality reduction, shown in Figure 5. Additionally, to quantitatively evaluate the effect of shift directions of different ICV methods, we calculate the following metrics. Given a query $\hat{x}$, we use $\boldsymbol{r}_{icl}$, $\boldsymbol{r}_{zs}$, $\boldsymbol{r}^*$ to denote the representation of the first answer token obtained by 32-shot ICL, Zero-Shot, specific ICV methods, respectively. Then we calculate the standard shift direction as $\boldsymbol{s}_{gt} = \boldsymbol{r}_{icl} - \boldsymbol{r}_{zs}$ and the shift direction of specific ICV as $\boldsymbol{s}^* = \boldsymbol{r}^* - \boldsymbol{r}_{zs}$. Finally, we define the shift direction similarity as the cosine similarity between $\boldsymbol{s}^*$ and $\boldsymbol{s}_{gt}$, indicating how closely the shift direction of the ICV method aligns with the standard shift direction. The results are presented in Table 4.

From Figure 5, we can find that 32-shot ICL exhibits a significant shift compared to Zero-Shot, visualizing the shift effects given in Eq. 4. Considering both Table 4 and 1, we find that the shift direction similarity has positive correlation to the accuracy: if a method has large direction similarity, it also has better performance. For example, among non-learnable methods, TV has higher shift direction similarity than other ones,

Table 4: The shift direction similarities of different ICV methods.

|  | TV | FV | PCA-ICV | LIVE |
|---|---|---|---|---|
| VQAv2 | 0.486 | -0.106 | 0.027 | **0.742** |
| OKVQA | 0.326 | 0.218 | -0.190 | **0.829** |

then it has better accuracy in Table 1. Furthermore, for LIVE which has the best accuracy in Table 1, its shift direction similarity is also the highest, which is 0.742/0.829 on VQAv2/OKVQA, validating that LIVE can produce shifts in query samples similar to 32-shot ICL, as visualized in Figure 5.

Table 5: Direct decoding of the different ICV methods.

| Methods | Decoding Top-10 Tokens of different methods in order of decreasing probability |
|---|---|
| TV | 'No', 'Yes', 'no', 'It', 'I', 'No', 'The', 'A', 'yes', 'Not' |
| FV | '.', 'in', ',', '(', 'for', 'and', '...', 'to', 'on', 'I' |
| PCA-ICV | 'none', 'there', 'no', 'the', 'not', 'None', 'dep', '_yes', 'unknown', 'yes' |
| LIVE | 'Question', '_Short', '?', 'no', 'QUEST', 'questions', '$?', 'answer', 'Short', '_questions' |

Table 6: The frequency of yes/no hallucinations and meaningless responses.

| | Zero-Shot | TV | FV | PCA-ICV | LIVE |
|---|---|---|---|---|---|
| yes/no Hallucination | 5 | 111 | 7 | 4 | **3** |
| Meaningless Answer in yes/no | 0 | 0 | 0 | 2 | **0** |
| Meaningless Answer in number | 2 | 0 | 2 | 522 | **0** |
| Meaningless Answer in other | 257 | 0 | 247 | 2072 | **2** |

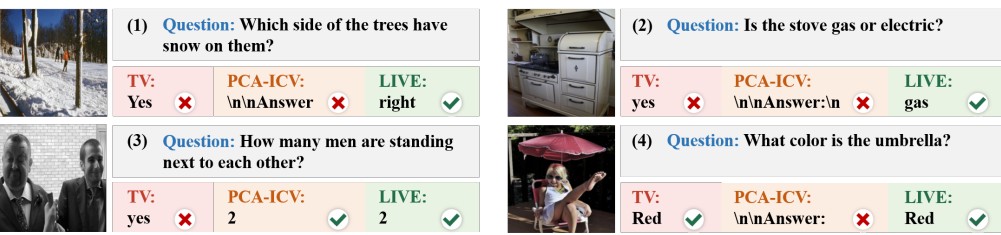

Figure 6: Visualizations of the cases where non-learnable methods appear yes/no hallucinations and meaningless responses.

Such positive correlation validates the effectiveness of our motivation that a single LIVE can indeed simulate the ICL capability of LMMs by shifting the direction of the query representation.

### 4.4.2 Why Non-Learnable Methods are Poor on VQA?

**Decoding ICV To Tokens:** We follow previous studies [49–51] to analyze the parameters of Transformers by directly decoding them into vocabulary tokens. Specifically, given a vector $v \in \mathbb{R}^{1 \times d}$, it can be projected using the unembedding matrix $\mathbf{E} \in \mathbb{R}^{d \times \mathcal{N}}$ of LMMs to obtain the corresponding token probability distribution $p$, where $\mathcal{N}$ is the vocabulary size:

$$p = \text{softmax}(v \cdot \mathbf{E}) = \frac{\exp(v \cdot \mathbf{E})}{\sum_j \exp(v \cdot \mathbf{E})_j}. \tag{8}$$

We calculate the $p$ of the vectors got from different methods in VQAv2 and select the top-10 tokens with the highest probabilities in $p$ shown in Table 5. We can see that the tokens got from FV are not highly relevant to the VQA task, which proves that FV does not capture the task information of VQAv2. On the other hand, the frequency of "yes" and "no" tokens is relatively high in the decoding results of PCA-ICV and TV, suggesting that they prefer to capture the simple patterns from demonstrations, *e.g.* yes/no, but struggle to grasp the overall task information of complex VQA. In contrast, the tokens of LIVE decoding are not biased to specific answers like yes/no, suggesting it abstracts more summary task knowledge of VQA.

**Hallucinations and Invalid Responses.**

VQA contains various answer types and for convenience, VQAv2 divides them into three categories: "yes/no", "number", and "other". After delving deeper into the answer details, interestingly, we find that TV frequently answers "yes or no" to number/other questions as shown in Figure 6 (1)(2)(3). We term this phenomenon the **yes/no hallucination** and count the frequency of the yes/no hallucination over all test data samples for different methods in Table 6. We can find that TV appears 111 times of yes/no hallucination, being consistent with the observations in Table 5, suggesting TV is biased to yes/no type question. We also observe that non-learnable methods tend to respond meaningless text (*e.g.* "\n") when responding to "number/other" questions as shown in Figure 6 (1)(2)(4). Table 6 shows the number of meaningless answers. We find that PCA-ICV and TV have more chance to return

meaningless answers for "other" questions, suggesting these methods do not capture the overall task information of VQA and are not able to answer some less frequently appeared questions. However, for LIVE, it has less yes/no hallucination and meaningless responses, validating that LIVE captures more robust task information of VQA.

## 5 Conclusion

To address the two major drawbacks of ICL in LMM—long computation time and sensitivity to demonstration selection—we try to apply non-learnable ICV methods from NLP to solve VQA. However, due to the complexity of VQA and the significant biases often inherent in non-learnable methods, the performance is unsatisfactory. Then we propose the Learnable ICV (LIVE) to overcome this drawback. By learning the general shift direction from a large amount of ICL data, LIVE successfully replaces the role of demonstrations in ICL. Experiments validate that LIVE outperforms traditional ICL methods and other non-learnable ICV methods on two VQA datasets. Experiments also show that LIVE, compared to LoRA, maintains excellent performance with minimal data, suggesting LIVE is a new research direction for LMMs to solve multimodal tasks. In the future, we will explore the application of LIVE on more multimodal tasks by various LMMs.

## Acknowledgement

This work is supported by National Science Foundation of China (62206048), Natural Science Foundation of Jiangsu Province (BK20220819), Young Elite Scientists Sponsorship Program of Jiangsu Association for Science and Technology Tj-2022-027, Fundamental Research Funds for the Central Universities(2242024k30035) and Big Data Computing Center of Southeast University

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

# A   Implementation Details.

## A.1   LIVE Hyperparameters

Table 7 presents the hyper-parameters utilized for training the LIVE. The optimizer denotes the optimization algorithm employed during model training. For $V$ and $\alpha$, we use different learning rate to optimize. The $\lambda$ represents the weight assigned to $\mathcal{L}_{gt}$ in Eq 7 during training. Precision refers to the float precision type used for model weights and gradient descent throughout the ICV training process. Weight Decay signifies the rate of weight decay applied during training, the warm up value is set to 0.01. while accumulate batches denotes the batch size for gradient accumulation during the training phase.

Table 7: VQAv2 and OKVQA LIVE Training Parameters

| Hyperparameter | VQAv2 | OKVQA |
|---|---|---|
| optimizer | AdamW [52] | AdamW |
| learning rate of $\alpha$ | 1e-2 | 1e-2 |
| learning rate of $V$ | 1e-3 | 5e-3 |
| $\lambda$ | 0.5 | 0.5 |
| weight decay | 1e-3 | 1e-3 |
| precision | FP16 | FP16 |
| batch size | 2 | 2 |
| warm up | 0.1 | 0.1 |
| accumulate batches | 8 | 8 |
| number of epochs | 10 | 10 |

## A.2   LoRA Hyperparameters

Table  8 details the hyper-parameters for the LoRA model trained during our experiment. Both the OKVQA and VQAv2 datasets use the same hyper-parameters.

Table 8: LoRA Training Parameters

| Hyperparameter | Value |
|---|---|
| optimizer | AdamW |
| learning rate | 1e-3 |
| LoRA matrix rank | 32 |
| LoRA dropout rate[53] | 0.05 |
| batch size | 2 |
| warm up | 0.1 |
| number of epochs | 10 |

## A.3   Inference and hardware details

In our ICV inference process, we employ the following hyperparameters: For ICV model inference, the maximum number of new tokens is set to 5, the number of beam searches is set to 3, the length penalty is set to 0, and the minimum number of generated tokens is set to 0. During the inference process, we utilize two Xeon Silver 3414 CPUs, one RTX 3090 GPU, and 384 GB of memory.

# B   Detailed Results

## B.1   The Detailed Inference Speed Experiments

This subsection provides a detailed presentation of some experimental data, primarily including forward propagation FLOPs and runtime cost, as well as a comparison of specific data between LIVE and LoRA. Table 9 presents the detailed results of the runtime and the FLOPs shown in Figure 3.

For a forward operation LIVE compared to $k$-shot ICL. The average token length during the forward inference is 38 tokens for Zero-Shot and LIVE, 107 tokens for 4-shot ICL, 187 tokens for 8-shot ICL, 330 tokens for 16-shot ICL, and 633 tokens for 32-shot ICL.

Table 9: Comparison of FLOPs and Runtime for LIVE and $k$-shot ICL

| Mertic | Zero-Shot | LIVE | 4-shot ICL | 8-shot ICL | 16-shot ICL | 32-shot ICL |
|---|---|---|---|---|---|---|
| FLOPs (TFLOPs) | 0.935 | 0.936 | 3.568 | 6.375 | 12.341 | 23.364 |
| Runtime (ms) | 56.69 | 56.81 | 92.44 | 158.21 | 266.13 | 468.55 |

## B.2 The Detailed Accuracy of Different Training Dataset

Table 10 presents a comparative analysis of the results between LoRA and LIVE results on different training dataset size in Section 4.3.

## B.3 Non-Learnable Methods Results

**Function Vector:** This section examines the test results of the function vector employed in the experiment across different layers of the IDEFICS-9B model. Table 11 presents the test results for VQAv2, and Table 12 shows the test results for OKVQA. The Best result of VQAv2 is 10th layer's result, which reaches 30.21, and the best result of OKVQA is 31.02 from the first layer.

Table 10: Accuracy (%) of LoRA and LIVE on different training set sizes

| Dataset | Method | Training set size | | | | | | | | | |
|---|---|---|---|---|---|---|---|---|---|---|---|
| | | 100 | 300 | 500 | 700 | 900 | 1000 | 2000 | 4000 | 6000 | 8000 |
| VQAv2 | LIVE | 29.43 | 49.66 | 49.93 | 51.03 | 53.42 | 53.71 | 56.64 | 57.76 | 58.33 | 58.54 |
| | LoRA | 29.52 | 39.02 | 40.01 | 40.99 | 41.99 | 42.81 | 45.26 | 47.71 | 48.51 | 49.02 |
| OKVQA | LIVE | 30.64 | 38.41 | 41.87 | 45.82 | 46.09 | 46.05 | 47.22 | 48.39 | 49.68 | 50.08 |
| | LORA | 30.37 | 13.38 | 20.01 | 23.42 | 25.01 | 25.66 | 29.52 | 33.08 | 34.17 | 34.21 |

Table 11: The Function Vector Accuracy (%) acorss differnt layers on VQAv2.

| VQAv2 | layer:1 | layer:2 | layer:3 | layer:4 | layer:5 | layer:6 | layer:7 |
|---|---|---|---|---|---|---|---|
| | 29.28 | 29.0 | 28.5 | 27.43 | 27.94 | 28.7 | 29.17 |
| | layer:9 | layer:10 | layer:11 | layer:12 | layer:13 | layer:14 | layer:15 |
| | 29.34 | 30.21 | 29.94 | 29.48 | 29.52 | 29.38 | 29.62 |
| | layer:17 | layer:18 | layer:19 | layer:20 | layer:21 | layer:22 | layer:23 |
| | 29.51 | 29.59 | 29.5 | 29.28 | 29.24 | 29.29 | 29.11 |
| | layer:25 | layer:26 | layer:27 | layer:28 | layer:29 | layer:30 | layer:31 |
| | 29.34 | 29.19 | 29.26 | 29.18 | 29.19 | 29.45 | 29.32 |

**Task Vector:** This section examines the test results of the task vector utilized in the experiment across various layers of the IDEFICS model. Table 13 displays the test results for VQAv2, while Table 14 shows the test results for OKVQA. The task vector is derived using 32 question-answer pairs. The best result of the task vector on VQAv2 is at the 10th layer, reaching 43.68, while the best result on OKVQA is at the 12th layer, reaching 32.68.

**PCA-ICV:** PCA-ICV employs a weighting factor $\alpha$ to regulate the degree of interference ICV has on the model. We test various values of $\alpha$ to assess performance. Table 15 shows the results of PCA-ICV on VQAv2 and OKVQA datasets with different $\alpha$ value and extract samples. It is evident

Table 12: The Function Vector Accuracy (%) acorss differnt layers on OKVQA

| OKVQA | layer:1 | layer:2 | layer:3 | layer:4 | layer:5 | layer:6 | layer:7 | layer:8 |
|---|---|---|---|---|---|---|---|---|
| | 31.02 | 30.57 | 30.23 | 30.27 | 30.04 | 30.36 | 30.61 | 30.23 |
| | layer:9 | layer:10 | layer:11 | layer:12 | layer:13 | layer:14 | layer:15 | layer:16 |
| | 30.56 | 30.62 | 30.6 | 30.27 | 30.22 | 30.19 | 30.2 | 30.3 |
| | layer:17 | layer:18 | layer:19 | layer:20 | layer:21 | layer:22 | layer:23 | layer:24 |
| | 30.44 | 30.25 | 30.27 | 30.17 | 30.22 | 30.16 | 30.31 | 30.23 |
| | layer:25 | layer:26 | layer:27 | layer:28 | layer:29 | layer:30 | layer:31 | layer:32 |
| | 30.42 | 30.32 | 30.34 | 30.25 | 30.3 | 30.34 | 30.4 | 30.28 |

Table 13: The Task Vector Accuracy (%) acorss differnt layers on VQAv2

| VQAv2 | layer:1 | layer:2 | layer:3 | layer:4 | layer:5 | layer:6 | layer:7 |
|---|---|---|---|---|---|---|---|
| | 28.18 | 28.82 | 30.47 | 30.53 | 36.41 | 35.35 | 36.17 |
| | layer:9 | layer:10 | layer:11 | layer:12 | layer:13 | layer:14 | layer:15 |
| | 41.72 | 43.68 | 40.33 | 34.32 | 16.91 | 15.42 | 14.53 |
| | layer:17 | layer:18 | layer:19 | layer:20 | layer:21 | layer:22 | layer:23 |
| | 12.94 | 12.44 | 12.28 | 11.89 | 11.98 | 11.78 | 11.44 |
| | layer:25 | layer:26 | layer:27 | layer:28 | layer:29 | layer:30 | layer:31 |
| | 12.8 | 12.95 | 12.97 | 12.86 | 12.87 | 13.29 | 14.7 |

Table 14: The Task Vector Accuracy (%) acorss differnt layers on OKVQA

| OKVQA | layer:0 | layer:1 | layer:2 | layer:3 | layer:4 | layer:5 | layer:6 | layer:7 |
|---|---|---|---|---|---|---|---|---|
| | 13.58 | 14.4 | 15.3 | 15.0 | 15.79 | 18.06 | 19.18 | 22.37 |
| | layer:8 | layer:9 | layer:10 | layer:11 | layer:12 | layer:13 | layer:14 | layer:15 |
| | 21.71 | 22.93 | 32.5 | 31.99 | 32.68 | 29.38 | 21.4 | 17.27 |
| | layer:16 | layer:17 | layer:18 | layer:19 | layer:20 | layer:21 | layer:22 | layer:23 |
| | 6.49 | 0.94 | 0.5 | 0.53 | 0.46 | 0.32 | 0.34 | 0.29 |
| | layer:24 | layer:25 | layer:26 | layer:27 | layer:28 | layer:29 | layer:30 | layer:31 |
| | 0.28 | 0.3 | 0.3 | 0.29 | 0.33 | 0.33 | 0.33 | 1.11 |

that the optimal alpha for the VQAv2 dataset significantly differs from that for OKVQA. The optimal result of PCA-ICV on VQAv2 is 34.75 when alpha is set to 1e-2, while the best result for PCA-ICV is 30.59 when alpha is set to 1e-5.

Table 15: The PCA-ICV Accuracy (%) acorss differnt alpha

| Dataset | Samples | Alpha | | | |
|---|---|---|---|---|---|
| | | 1e-2 | 1e-3 | 1e-4 | 1e-5 |
| VQAv2 | 32 | 34.75 | 30.95 | 30.06 | 30.0 |
| OKVQA | 32 | 22.46 | 30.06 | 30.23 | 30.59 |

# C   General LIVE

LIVE is essentially a shift vector, allowing us to control the shift directions of query representations, which means we can control the shift direction of LMMs by adding or subtracting different shift vectors. Therefore, we can use several LIVE trained on different VQA dataset to get the **General LIVE**. Specifically, given the diverse VQA task set $T = \{t_1, \ldots, t_n\}$, where $t_i$ is a VQA task and $n$ is the number of tasks, we train the task-specific LIVE $V^i = \{v_1^i, \ldots, v_L^i\}$ and the weight factor $\alpha^i = \{\alpha_1^i, \ldots, \alpha_L^i\}$ on each VQA dataset. Then, we have the LIVE set $\mathcal{V}_{set} = \{V^1, \ldots, V^n\}$ and its weight factors set $\alpha_{set} = \{\alpha^1, \ldots, \alpha^n\}$. The General LIVE has the same shape of task-specific LIVE. For the $l$-th layer, it is defined as $v_l = \sum_i \alpha_l^i v_l^i$. During inference, we use the vector $v_l$ to shift the original representation, resulting in $\mathcal{M}_l(\hat{x}_i)' = \mathcal{M}_l(\hat{x}_i) + v_l$. We average the LIVE trained in OKVQA and VQAv2 to get the general ICV and evaluate the performance of the general ICV $V_g = \{v_1, \ldots, v_L\}$ on OKVQA and VQAv2, with the results presented on Table 16.

Table 16: Accuracy (%) for 32-shot ICL, task-specific LIVE, General LIVE.

| Methods | 32-shot ICL | LIVE | General LIVE |
|---------|-------------|------|--------------|
| VQAv2   | 56.18       | 58.54 | 56.17        |
| OKVQA   | 48.48       | 50.08 | 49.52        |

The results indicate that while the performance of the general ICV is somewhat reduced compared to task-specific LIVE; it decreased by 2.37 on VQAv2 and by 0.56 on OKVQA. However, its performance is very close to that of 32-shot ICL. Most importantly, it offers significant advantages in real-world scenarios where the distribution of test data is unknown. This makes the general ICV more suitable for practical applications, providing a robust solution that can be effectively utilized across varying environments. Furthermore, the findings highlight the scalability of LIVE: whenever a new VQA dataset is used to train LIVE, we only need to simply recalculate the mean shift vectors of all previously trained task-specific LIVEs and the new LIVE to generate a new general ICV, thus creating a more general VQA LIVE. This approach not only simplifies the adaptation process for diverse tasks but also ensures that the model maintains a high level of performance across different applications. The results, therefore, underscore the potential of general ICVs to enhance the flexibility and applicability of VQA systems in real-world settings.

# D   More exploratory experiments.

## D.1   Generalization Ability

We conducted supplementary experiments on other vision-language tasks and LMMs to demonstrate the general applicability of our LIVE method.

### D.1.1   Task Generalization

In Table 17, we utilized IDEFICSv1-9B as the baseline model to evaluate the performance of LIVE against ICL and LoRA on the COCO caption dataset[54], OcrVQA dataset[55], and VL-ICL dataset[56]. Experimental results demonstrate that our method consistently maintains advantages over both ICL and LoRA across various tasks.

Table 17: Accuracy (%) for 32-shot ICL, LoRA, and LIVE in various dataset.

| Task (Metric)       | 32-shot ICL | LoRA   | LIVE       |
|---------------------|-------------|--------|------------|
| COCO (CIDEr)        | 106.31      | 109.18 | **117.38** |
| VL-ICL Textocr (ACC)| 21.5        | 22.5   | **24.0**   |
| VL-ICL Clevr (ACC)  | 33.5        | 35.5   | **37.0**   |
| OcrVQA (ACC)        | 16.3        | 15.9   | **17.5**   |

### D.1.2 Model Generalization

To test the generalizability of our method across different architectural models, we extended LIVE to IDEFICSv2 model[57]. It is noteworthy that IDEFICSv2 (8B-base) differs from IDEFICSv1 in its handling of image-text interactions; while IDEFICSv1 uses a cross-attention mechanism for the fusion of image and text tokens, IDEFICSv2 utilizes an additional projection layer to map image information into the text space for LMM processing. These two architectures represent the primary approaches within LMM. Conducting experiments with this model can validate the generalizability of LIVE.

Due to the longer input sequence of IDEFICSv2 compared to IDEFICSv1, we reached the limits of our GPU memory during 8-shot ICL inference. Therefore, we compared the LIVE trained on 8-shot data with 1-8 shot ICL. As shown in Table 18, LIVE improves upon 8-shot ICL by 6.1% and 0.84% on VQAv2 and OKVQA, respectively. In terms of inference speed, it surpasses 8-shot ICL by a factor of 8.95. These results demonstrate that the LIVE method is both generalizable and effective within LMM, excelling in both inference accuracy and speed compared to ICL.

Table 18: The performance comparison between LIVE and ICL on IDEFICS-v2-8B.

| Method | VQAv2 ACC | OKVQA ACC | Inference Time | FLOPs (TFLOPs) |
|---|---|---|---|---|
| Zero-shot | 55.39 | 43.08 | 0.087 ($\times$ 0.97) | 3.110 ($\times$ 0.99) |
| 1-shot ICL | 60.33 | 45.65 | 0.168 ($\times$ 1.90) | 6.497 ($\times$ 2.07) |
| 2-shot ICL | 63.49 | 50.40 | 0.237 ($\times$ 2.68) | 9.610 ($\times$ 3.06) |
| 4-shot ICL | 64.88 | 53.18 | 0.395 ($\times$ 4.46) | 17.139 ($\times$ 5.46) |
| 8-shot ICL | 66.20 | 57.68 | 0.792 ($\times$ 8.95) | 32.130 ($\times$ 10.23) |
| 8-shot LIVE | **70.30** | **58.52** | 0.089 ($\times$ 1) | 3.141 ($\times$ 1) |

### D.2 Performance with more shot

To investigate the impact of a higher number of shots on LIVE, we conducted a comparison with ICL at 48 and 64 shots. Detail results are shown in Table 19 .The results indicate that while increasing the number of shots improves the performance of both methods, they have reached a performance plateau. Notably, the gains achieved by LIVE remain superior to those of ICL. For example, with 64 shots LIVE achieves a 2.99%/1.85% improvement over ICL on VQAv2/OKVQ. Thus, although using more ICDs improves ICL performance, LIVE has a large improvement

Table 19: The performance comparison of ICL and LIVE with more shots.

| Benchmark | Method | Shot 16 | Shot 32 | Shot 48 | Shot 64 |
|---|---|---|---|---|---|
| VQAv2 | ICL | 55.7 | 56.18 | 56.67 | 56.71 |
|  | LIVE | **58.27** (+2.57) | **58.54** (+2.36) | **59.07** (+2.4) | **59.70** (+2.99) |
| OKVQA | ICL | 47.7 | 48.48 | 48.68 | 48.60 |
|  | LIVE | **49.71** (+2.01) | **50.08** (+1.6) | **50.55** (+1.87) | **50.45** (+1.85) |

### D.3 Shared layer LIVE

We conducted experiments to investigate the impact of using a shared bias vector across all layers (as shown in Table 20), and we observed a significant performance drop compared to LIVE: 20.6% on VQAv2 and 9.01% on OKVQA. This performance drop aligns with the findings in [24] that different Transformer layers play diverse roles in ICL. Also, Eq 5 show that LIVE vectors are layer-specific. Then using a single shared vector fails to capture the distinct information processed at each layer, leading to the observed decline in accuracy.

### D.4 Detail comparisopn of LoRA

To provide a more detailed comparison with LoRA, we made the following modifications to LoRA. First, based on the loss function formula in Eq. 7 we applied the same loss function to LoRA. As shown in Table 21, even after modifying the loss function, LoRA's performance remains significantly lower

Table 20: The performance comparison of using a shared LIVE across all layer (Share LIVE) with LIVE and ICL

| Method | VQAv2 | OKVQA |
|---|---|---|
| 32-shot ICL | 56.18 | 48.48 |
| 32-shot LIVE | **58.54** | **50.08** |
| 32-shot Share LIVE | 37.94 (-20.6) | 41.07 (-9.01) |

than LIVE on both VQAv2 (49.48% vs. 58.54%) and OKVQA (37.02% vs. 50.08%). Additionally, we fine-tuned LoRA with more parameters, but similarly, increasing the parameter count did not close the performance gap between LoRA and our LIVE, further demonstrating the effectiveness of LIVE's unique design.

Table 21: The performance comparison of the different implementations of LoRA with LIVE. "LoRA" integrates LoRA into the token classification head of the last layer to minimize the trainable parameters, "LoRA with KL" uses 32-shot ICDs and applies the same KL loss as our LIVE, and "LoRA O_proj all layers" integrates LoRA into the O_proj of all layers.

| Method | VQAv2 | OKVQA | Parameters Number |
|---|---|---|---|
| LoRA | 49.02 | 34.21 | 1.15M (x 8 times) |
| LoRA with KL | 49.48 | 37.02 | 1.15M (x 8 times) |
| LoRA O_proj all layers | 57.57 | 45.03 | 10.5M (x 80 times) |
| LIVE | **58.54** | **50.08** | 0.13M (x 1 time) |

### D.5 Comparison between untrained LIVE

We evaluated the impact of untrained LIVE by initializing them randomly with a normal distribution and tested on VQAv2 (Table 22), revealing that untrained LIVE has similar results as zero-shot, which are largely worse than the trained LIVE. We also visualized the embeddings using t-SNE to see the shift effect in Fig 7 (A), showing that the shift effect of untrained LIVE is similar to zero-shot ICL, while learned LIVEs align more closely with 32-shot ICL, proving that the training process is important.

Table 22: The performance comparison of untrained LIVE, trained LIVE, ICL, and Zero-shot.

| Task | Zero Shot | Untrained LIVE | 32-shot ICL | Trained LIVE (Paper) |
|---|---|---|---|---|
| VQAv2 | 29.25 | 30.3 | 56.18 | **58.54** |
| OKVQA | 30.54 | 30.2 | 48.48 | **50.08** |

Table 23: The performance comparison between LIVE and diverse ICL methods.

| Task | Method | Shot 1 | Shot 4 | Shot 8 | Shot 16 | Shot 32 | Average |
|---|---|---|---|---|---|---|---|
| VQAv2 | LIVE | 56.84 | 57.60 | 58.25 | 58.27 | 58.54 | **57.90** |
| | ICL | 51.39 | 53.72 | 54.24 | 55.70 | 56.18 | 54.25 |
| | RICE | 47.83 | 53.54 | 55.04 | 56.89 | 58.07 | 54.27 |
| | MMICES | 48.22 | 54.99 | 56.16 | 57.02 | 57.98 | 54.87 |
| OKVQA | LIVE | 47.51 | 47.68 | 49.4 | 49.71 | 50.08 | **48.88** |
| | ICL | 40.75 | 46.11 | 46.79 | 47.70 | 48.48 | 45.97 |
| | RICE | 40.67 | 47.07 | 48.92 | 50.73 | 51.11 | 47.70 |
| | MMICES | 40.39 | 47.13 | 50.19 | 50.46 | 50.61 | 47.76 |

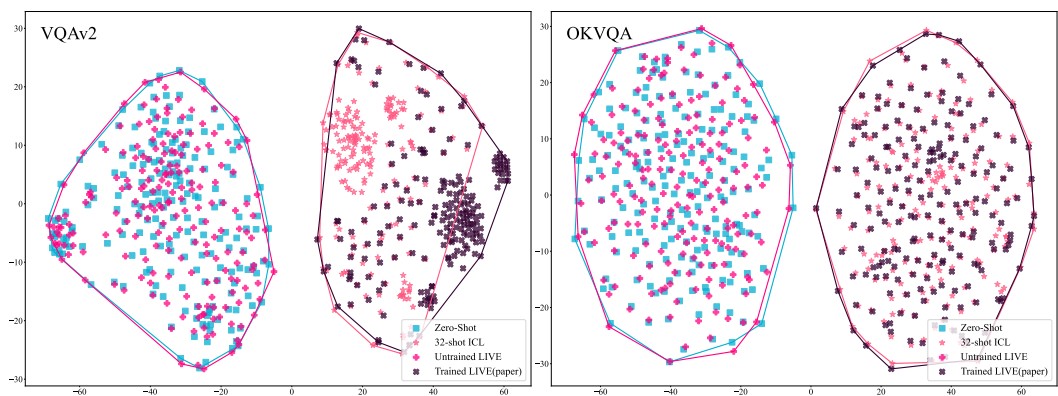

Figure 7: T-SNE visualization of first answer token over 200 queries. Left/right corresponds to VQAv2/OKVQA, respectively.

## D.6 Comparison between LIVE and diverse ICL methods.

In previous works[8, 19, 18], the importance of selecting ICDs for ICL in VQA and other vision-language tasks has been demonstrated. To reasonably compare the performance differences between LIVE and ICL, we adopted two methods for selecting ICL samples: RICE[43], which retrieves samples based on image similarity to the query, and MMICES[58], which combines both language and image similarity for ICD retrieval. The results comparing LIVE and ICL with different ICD selection strategies are presented in the Table 23, showing that even with optimized ICD selection, ICL still lags behind LIVE in terms of accuracy.

We also tried to use RICE samples to learn LIVE. However, the results were unsatisfactory, with only 57.68% accuracy on VQAv2, while using random samples to train LIVE got 58.54% accuracy. This may be because using RICE samples will make each training input sequence contain abundant individual characteristics, increasing the difficulty for LIVE to learn common task knowledge, and even causing LIVE to learn spurious correlations between the similar ICDs and the query.

