# OpenReview forum: "LIVE: Learnable In-Context Vector for Visual Question Answering"
_NeurIPS.cc/2024/Conference — NeurIPS 2024 poster_

### Official Review · Reviewer_BDyQ · 2024-07-08

**Soundness:** 2
**Presentation:** 3
**Contribution:** 2
**Rating:** 5
**Confidence:** 4

**Summary:**

This paper aims to improving the in-context learning performance of multimodal models on VQA tasks. The paper proposes a Learnable In-Context Vector (L-ICV) method to distill essential task information from demonstrations into a single vector, reducing computational costs and enhancing accuracy in VQA tasks. Experiments show that L-ICV can achieve better ICL performance compared to random 32-shot demonstrations and significantly reduce the computation cost.

**Strengths:**

The topic is important for the community, and the proposed method is an insightful exploration that can be beneficial. The overall writing of this paper is clear and easy to follow. L-ICV reduces computational costs, requiring only a fraction of the FLOPs and inference time compared to traditional ICL methods.

**Weaknesses:**

One of the major weaknesses is that the evaluation of L-ICV is limited to a small set of models and datasets. Only one model (IDEFICS-9B) and two datasets (VQAv2, OKVQA) are reported across the paper.  This limitation raises questions about the generalizability of the method to other multimodal models and different dataset configurations. Besides, recent literature has discussed the in-context limitation of these datasets [1, 2, 6, 7] and a broader comparison of tasks that rely more on visual in-context learning ability should be beneficial to showcase the advantage of L-ICV.

The second major issue is the relatively limited increase of the ICL performance. Although the L-ICV method demonstrates improved efficiency and performance, the reported increase in accuracy is not particularly significant. while L-ICV has been trained on a large dataset, the expected performance gains are not as high as one might anticipate given the extensive training resources utilized.

Additionally, the paper only compares L-ICV with ICL using randomly chosen demos and does not compare L-ICV with the classical RICE (Retrieval-based In-Context Examples) method [3, 4, 5] or a recently proposed MMICES [6]. RICES and MMICES samples can lead to a direct ICL performance boost, and a direct comparison would help to position L-ICV within the current state-of-the-art. Besides, if L-ICV is trained using RICES samples rather than randomly sampled demos, will the performance be better? This should be an interesting exploration that should be conducted.

Furthermore, the in-context learning ability of large models is practical and useful as it does not need any parameter optimizations and only requires a handful of demonstrations to quickly adapt to new tasks. To enhance this ability, it would be better if the proposed method also requires minimal dataset, tuning, and optimization effort to achieve similar or better performance.

Overall, while L-ICV presents a valuable exploration of the field, addressing these weaknesses through broader evaluations, detailed statistical analyses, comprehensive comparisons, and further optimization efforts would enhance the method's credibility and impact.

[1] Zong, Yongshuo, Ondrej Bohdal, and Timothy Hospedales. "VL-ICL Bench: The Devil in the Details of Benchmarking Multimodal In-Context Learning." *arXiv preprint arXiv:2403.13164* (2024).

[2] Doveh S, Perek S, Mirza M J, et al. Towards multimodal in-context learning for vision & language models[J]. arXiv preprint arXiv:2403.12736, 2024.

[3] Alayrac, Jean-Baptiste, et al. "Flamingo: a visual language model for few-shot learning." *Advances in neural information processing systems* 35 (2022): 23716-23736.

[4] Awadalla, et al. "Openflamingo: An open-source framework for training large autoregressive vision-language models." *arXiv preprint arXiv:2308.01390* (2023).

[5] Yang, et al. "An empirical study of gpt-3 for few-shot knowledge-based vqa." *Proceedings of the AAAI conference on artificial intelligence*. Vol. 36. No. 3. 2022.

[6] Chen, et al. "Understanding and Improving In-Context Learning on Vision-language Models." *arXiv preprint arXiv:2311.18021* 1.2 (2023).

[7] Baldassini, Folco Bertini, et al. "What Makes Multimodal In-Context Learning Work?." *Proceedings of the IEEE/CVF Conference on Computer Vision and Pattern Recognition*. 2024.

**Questions:**

Please see the Weaknesses section.

**Limitations:**

The limitations of this paper are threefold.

1. The experiments are relatively limited regarding the model types and evaluation tasks.
2. The proposed method requires extra training, optimization, and hyperparameter tuning effort. This added complexity could be a barrier to adoption, especially when there is only a limited set of demo data.
3. The increased performance is not significant, especially considering the training and tuning effort. This raises questions about the practical benefits of the proposed method relative to the effort involved in its implementation.

---

> ### Author Rebuttal · Authors · 2024-08-07
>
> **W1: Generalizability of L-ICV**
>
> We also follow your suggestion to test L-ICV on the VL-ICL Benchmark[A] and the results are given in Table C. Due to the rebuttal's time constraints, we only tested two subtask performances in the VL-ICL Benchmark. We also tested L-ICV on IDEFICS v2 and some new tasks to validate the generalizability of L-ICV in Table C&E and please refer to **"W2\&W3" of R.fRTc** for more details.
>
> [A] VL-ICL Bench: The Devil in the Details of Benchmarking Multimodal In-Context Learning.
>
> **W2: Performance Improvements**
>
> It should be stressed that the major motivation of ICV is to accelerate the inference speed instead of improving the ICL performance. To train L-ICV, it costs us 11 $\times$ 3090 GPU hours on  8000 VQAv2 samples, occupying 1.8% of the VQAv2 dataset, which means that the total cost is not intolerable. Notably, all ICDs are randomly sampled from 8000 samples. After training, the inference speed can be largely improved and please refer to **"Q1" of R.8z8x** to see the trade-off between reduced inference time and additional training time.
> Furthermore, if L-ICV is trained by 1-shot ICD, the cost will be much less, 1$\times$ 3090 GPU hours on 8000 samples, while 1-shot L-ICV achieves 5.45%/6.76% higher accuracy than 1-shot ICL on VQAv2/OKVQA, which is a significant improvement. In this case, even 1-shot L-ICV achieves better results than 32-shot ICL on VQAv2 and 16-shot ICL on OKVQA. In the 32-shot case, the improvement of L-ICV is less significant, while one reason is that modern LMM with ICL ability can not deal with long input sequences. Specifically, these LMMs are trained by 5-shot ICEs, meaning limited capacity for more shot inputs [6,7].
> We also compared L-ICV with the LoRA method requiring training, and L-ICV consistently outperformed LoRA across various training set sizes, indicating that not all training-required methods achieve such good performance.
>
> **W3.1: Comparison with RICES and MMICES**
>
> Retrieving ICDs based on query sample and learning an ICV represent two major but orthogonal directions in ICL. The former exploits more individual characteristics of the query sample to improve the ICL performance. While the latter exploits more common knowledge to get a single intervene vector to improve the ICL inference speed. Considering their different motivations, we do not directly compare them in the submitted manuscript. Here we follow your suggestion to discuss their relative advantages and disadvantages.
> 1.  Table H compares L-ICV with RICE/MMICES, showing that on average, L-ICV is better than RICE/MMICES on VQAv2/OKVQA, while be worse than RICE/MMICES on OKVQA when using more-shot ICDs since this tasks asks for more external knowledge and RICE/MMICES can provide such knowledge when returning similar ICDs.
> 2. RICE and MMICES require additional image and text encoders to retrieve ICDs from a database, occupying computational resources and causing significant latency during inference. Moreover, RICE and MMICE require LMMs to deal with many-shot ICDs, thus requiring longer inference time. To sum up, it is less favorable for real-time deployment than our L-ICV.
> 3. RICE and MMICES retrieve ICDs based on feature similarity, which makes LMM produce more short-cut learning as demonstrated in [16,17]. This is because that LMM tends to mimic the text description of the most similar samples and then neglect the vision contents. However, since L-ICV tries to capture common knowledge from diverse combinations of ICDs, it is less affected by short-cut learning.
>
> **W3.2: Using RICE samples to learn L-ICV**
>
> Actually, at the beginning, we also tried to use RICE samples to learn L-ICV. However, the results were unsatisfactory, with only 57.68% accuracy on VQAv2, while using random samples to train L-ICV got 58.54% accuracy. This may be because using RICE samples will make each training input sequence contain abundant individual characteristics, increasing the difficulty for L-ICV to learn common task knowledge, and even causing L-ICV to learn spurious correlations between the similar ICDs and the query. We will add this experiment and discussion in the revision.
>
> **W4: Why update parameters in a parameter-efficient way.**
>
> Indeed, the advantage of ICL is that it does not require parameter updates, but its sensitivity to ICDs is a drawback. Therefore, many papers attempt to train auxiliary models to select or order ICDs for improving ICL performance [10, 11]. Our L-ICV follows this research direction but uses less data and updates fewer parameters than other parameter-efficient methods like LoRA.
> Moreover, due to the sequence length inference issue, ICL faces significant latency during deployment when applying some retrieval-based methods. However, L-ICV achieves more stable and superior performance with minimal training, striking a trade-off between PEFT and ICL, combining their strengths to enhance inference performance and reduce latency.

---

> > ### Comment · Reviewer_BDyQ · 2024-08-08
> > **Two more questions**
> >
> > Thank you for the feedback.
> >
> > I understand that the main goal of this paper is to accelerate inference time. I agree that learning ICVs and retrieving ICDs based on a query can be orthogonal. I also appreciate the newly added experimental results and findings during training. I have two further questions below.
> >
> > In a realistic setting, where a new task has only a few demonstrations and no large annotated dataset for training, how can LICV be better utilized? I believe this is more practical compared to VQAv2 with 8000 samples in applications. Also, how many images are used as training data for the VL-ICL Clevr and Textocr tasks?
> >
> > Additionally, I am curious about the transferability of the context vector trained on one task's data to another relatively similar dataset, such as being trained on VQAv2 but tested on OKVQA. Do you have the experimental results and analysis regarding this matter?

---

> ### Author Response · Authors · 2024-08-13
>
> We appreciate your response. We conducted additional experiments to show how L-ICV performs with a few demonstrations and to explore its transferability. We also include more discussion of these questions.
>
> **Q1: A few demonstrations to train L-ICV**.
>
> L-ICV only uses 800 samples (the total number of samples in the training split) in the VL-ICL Clevr and Textocr tasks.
>
> To further explore how much data L-ICV requires to be effective for those tasks, we conducted additional experiments. Theoretically, even with a small number of ICD samples, they can generate a large number of ICD sequences. For example, there are  $A_{300}^{32}$ ICD sequences for 300 data samples. Therefore, by extending the training steps of L-ICV, the model can learn from more randomly arranged samples. We trained for 5 additional epochs on the original basis.
> In the TextOcr task, only 50 samples were needed to achieve **an accuracy of 27.5% (3.5% higher than 32-shot ICL)**. In the Clevr task, 200 samples were required to achieve **an accuracy of 34.5% (1% higher than 32-shot ICL)**．Additionally, we conducted further tests on VQAv2. Due to time constraints, we used 4-shot training on 300 samples for L-ICV. After 15 epochs, L-ICV achieved **an accuracy of 53.75% (0.23% higher than 4-shot ICL)**. These results indicate that L-ICV can be effective even with a relatively small amount of data. Given time, as we continue to increase the number of ICD sequences and training epochs, we may get satisfactory L-ICVs with fewer samples.
>
> **Q2: The transferability for L-ICV.**
>
> We followed your good suggestion to evaluate the transfer performance of L-ICV and ICL.  Specifically, for ICL, we used VQAv2/OKVQA samples as ICDs during inference when testing on OKVQA/VQAv2 (VQAv2->OKVQA/OKVQA->VQAv2). Similarly, for L-ICV, we used L-ICV trained on VQAv2/OKVQA for inference on OKVQA/VQAv2 (VQAv2->OKVQA/OKVQA->VQAv2). The results are shown in the table below:
>
> | Methods | VQAv2->VQAv2 | OKVQA->OKVQA | OKVQA->VQAv2 | VQAv2->OKVQA |
> |---------|--------------|--------------|--------------|--------------|
> | ICL     | 56.18        | 48.48        | 50.45        | 45.02        |
> | L-ICV   | **58.54**    | **50.08**    | **51.73**    | **45.98**    |
>
>
> It can be seen that L-ICV achieves better transferability compared to ICL. For example, in the OKVQA->VQAv2 experiment, L-ICV outperformed ICL by 1.28%, and in the VQAv2->OKVQA experiment, L-ICV outperformed ICL by 0.96%. One possible reason is that when using ICL, the individual characteristics of the ICDs from another task may have more damage than the general task summary of another task, which is captured by L-ICV. This demonstrates that the L-ICV we trained can be directly transferred to similar tasks while maintaining relatively good performance. It can also be found that the transferred L-ICV is worse than the original L-ICV, e.g., OKVQA->VQAv2 L-ICV is worse than VQAv2->VQAv2  L-ICV. This aligns with intuition, because when using samples from a task to train L-ICV, L-ICV will learn the Task Learning ability (refer **Q1&W1 of R.VKDo**) and learn the common knowledge of the task, which may have a distribution shift with anohter task and lead to the performance decline in other task.
>
> To compensate this effect, we further propose a general ICV, it enables us to combine knowledge from multiple datasets. As shown in Appendix C, Table 16, we simply evaluated that by using the mean vector of L-ICV trained on VQAv2 and OKVQA, and it can achieve results on both datasets that are comparable to 32-shot ICL. We believe this could be a promising direction for future research, as a general ICV learned from similar tasks could effectively address scenarios where a new task has only a few demonstrations and no large annotated dataset.
>
> However, it is out of scope of the discussion of this paper and we hope to leave it as future work.

---

> > ### Comment · Reviewer_BDyQ · 2024-08-13
> >
> > Dear Authors,
> >
> > Thanks again for the newly added experiments and analysis. It is good to see the L-ICV can also attain comparable or slightly better performance even with a small set of training samples. Also thanks for analyzing the transferability capacity of the learned context vectors. It is recommended that the authors add these two parts to the paper for a more thorough version.
> >
> > I am glad the authors have addressed most of my concerns and I am happy to increase my score by 1.
> >
> > Best

---

> > > ### Author Response · Authors · 2024-08-14
> > >
> > > Thank you for the insightful comments and constructive suggestions you provided during the review process. Your expertise and attention to detail have significantly enhanced the quality of our manuscript. We greatly appreciate the time and effort you invested in helping us improve our work.

---

### Official Review · Reviewer_fRTc · 2024-07-09

**Soundness:** 2
**Presentation:** 3
**Contribution:** 2
**Rating:** 4
**Confidence:** 3

**Summary:**

This paper proposes a method by introducing the idea of In-Context Vector in the field of NLP, by introducing L-ICV with a smaller number of parameters in LMM to replace in-context demonstrations to introduce external knowledge, and obtain through training The appropriate shift direction is combined in the query. By comparing with some existing methods, the author proves that L-ICV can significantly reduce computational costs while avoiding the bias in the selection of in-context demonstrations and improving the accuracy of VQA tasks.

**Strengths:**

(1) The paper is clearly written and easy to comprehend.

(2) The motivation for the proposed method is reasonable. Existing ICL methods are difficult to achieve good results while saving computing resources. Using a shifting vector instead of in-context demonstrations can reduce the computational cost while ensuring the effect.

**Weaknesses:**

(1) In the comparison of Section 4.2.1, only the inference time consumption of L-ICV and k-shot ICL is listed. How about other methods like TV, FV, etc.?

(2) The method's generalizability is questionable as it has only been tested on the IDEFICS model, with no consideration of other models.

(3) The paper only proves the effect of L-ICV on VQA tasks, which cannot be extended to other task scenarios and has limited application scope.

(4) In the ablation study, there is a lack of a control group using untrained L-ICV to prove the effectiveness of shift direction.

**Questions:**

(1) Why 32-shot demonstrations are chosen in L-ICV training? According to Table 3. As the Number of Demonstrations k increases, the performance gap between L-ICV and ICL gradually narrows. When k continues to increase, will the performance of ICL exceed that of L-ICV?

(2) Why is there a need for a unique shift effect for each layer in the L-ICV method, but in the comparative experiment of LoRA, only the LoRA module is added to the token classification head of the last layer?

(3) In the training stage of L-ICV, the k demonstrations used for training are also randomly selected. How to prove that this can eliminate one of the major challenges when applying ICL: the performance is sensitive to the selection of ICDs.

**Limitations:**

The authors have adequately addressed the limitations in the PaperChecklist.

---

> ### Author Rebuttal · Authors · 2024-08-06
>
> **W1: Inference Time Comparison of TV and FV**
>
> TV, FV and L-ICV have similar inference time. Specifically, TV takes 57.14 ms, FV takes 57.08 ms, L-ICV takes 56.81 ms for one sample, and Zero-Shot takes 56.69 ms. This similarity in inference times is because TV and FV replace the ICDs with a single vector to intervene in the model’s forward process, just like our L-ICV. However, L-ICV significantly surpasses TV and FV in terms of performance, achieving higher accuracy while also reducing inference costs. We will include these in the final.
>
> **W2: Generalizability of the Method**
>
> To test the generalizability, we extended L-ICV to IDEFICS v2-8B [A]. Notably, IDEFICS v2 differs in architecture from IDEFICS v1 (used in the paper). Currently, there are two main LMM architectures for ICL: (1) Models like IDEFICS v1 that use cross-attention to fuse image and text tokens, and (2) Models like IDEFICS v2 that employ additional projection layers to concatenate image and text tokens. Testing L-ICV on IDEFICS v2 can further demonstrate the generalizability across architectural designs.
> Since IDEFICS v2 concatenates image and text tokens, it has a longer input sequence than IDEFICS v1 when the same image-text pair is used. For instance, an image is represented by 64 tokens, resulting in 512 vision tokens in an 8-shot scenario, which pushed our GPU to its computational limits. Consequently, we trained L-ICV with an 8-shot ICD sequence and compared it with 1-8 shot ICL. The results in Table E show that L-ICV outperforms 8-shot ICL, achieving 4.1%/0.84% higher accuracy on VQAv2/OKVQA, respectively. Additionally, L-ICV is 8.95 times faster in inference speed compared to 8-shot ICL on IDEFICS v2.
>
> [A] What matters when building vision-language models?
>
> **W3: L-ICV on Other Tasks**
>
> Note that L-ICV is inherently task-agnostic as Equations (1-4) in Sec.3 can be used among various tasks.  We initially focused on VQA because it covers a wide range of vision-language challenges (Lines 56-59), which is a more challenging task for FV/TV. We follow your suggestion to test L-ICV on image captioning, VL-ICL Benchmark and OcrVQA (Table C). It can be found that L-ICV achieves 11.07 higher CIDEr than 32-shot ICL on IC, 3.5 higher ACC than 32-shot ICL on VL-ICL-clevr, 2.5 higher ACC than 32-shot ICL on VL-ICL-textocr, 1.2 higher ACC than 32-shot ICL on OcrVQA, validating the generalizability of L-ICV on various tasks.
>
>
> **W4: Ablation with Untrained L-ICV**
>
> We evaluated the impact of untrained L-ICV by initializing them randomly with a normal distribution and tested on VQAv2 (Table F), revealing that untrained L-ICV has similar results as zero-shot, which are largely worse than the trained L-ICV. We also visualized the embeddings using t-SNE to see the shift effect (Figure A), showing that the shift effect of untrained L-ICV is similar to zero-shot ICL, while learned L-ICVs align more closely with 32-shot ICL, proving that the training process is important.
>
> ---
>
>
> **Q1: Why do we choose 32-shot demonstrations and the performance gap with more shots?**
>
> The choice of 32-shot demonstrations is influenced by the LMMs’ capacity for understanding long texts, rather than by limitations of our method. Many related studies [6,7] typically use up to 32 shots due to these practical constraints.
> To investigate the impact of more-shot ICDs, we evaluated L-ICV with 48/64-shot ICDs (Table G), showing that while adding more shots improves performance for both methods, the relative gain for L-ICV remains higher than ICL. For example, with 64 shots, L-ICV achieves a 2.99%/1.85% improvement over ICL on VQAv2/OKVQA. Thus, although using more ICDs improves ICL performance, L-ICV has a larger improvement.
>
> **Q2: Why unique L-ICV for each layer and the comparable design for LoRA?**
>
> Since [22] finds that each layer plays distinct roles during ICL inference (Lines70-72), we use unique shift vectors for each layer where Equations (3-4) are applied to each single layer, making L-ICV layer-specific. Using layer-specific L-ICV requires 32 $\times$ 4096 parameters. We integrated LoRA into the token classification head of the last layer to minimize the trainable parameters (still 8 times more parameters than L-ICV).
> To ensure a comprehensive comparison, we also evaluated LoRA when applied to the `O_proj` of all layers, as shown in Table B. This configuration results in an 80-fold increase in parameters, allowing LoRA to achieve decent performance but still falling short of L-ICV by 0.97% in VQAv2 and 5.05% in OKVQA, suggesting that L-ICV is more effective than LoRA.
>
> **Q3: How to address the performance sensitivity to ICD selection with L-ICV?**
>
> As discussed in Lines 67-70, it is precisely because we aimed to reduce the sensitivity of ICD that we randomly select 32-shot examples during training. Generally, each ICD has individual characteristics and some common knowledge about the task, while the ICL sensitivity stems from the individual characteristics. To encourage the ICV to learn common knowledge, during training, each in-context sequence is set to a random combination of ICDs, which forces ICV to capture the most common knowledge while disregarding these individual characteristics. Please refer to **"Q1 & W1" of R.VKDo** for more details.

---

### Official Review · Reviewer_8z8x · 2024-07-10

**Soundness:** 4
**Presentation:** 3
**Contribution:** 3
**Rating:** 8
**Confidence:** 4

**Summary:**

The study introduces a Learnable In-Context Vector (L-ICV) for Visual Question Answering (VQA) tasks. This approach extracts task information from demonstrations to improve the performance of Large Multimodal Models (LMMs) while reducing computational costs.

**Strengths:**

* The writing of this paper is excellent, the text is fluent, and the motivation is easy to understand.
* The proposed method is novel and effective, significantly reducing the cost of reasoning.
* The experiment is very solid, and the statistics in Figure 5 are impressive, intuitively showing the offset of the in-context vector.

**Weaknesses:**

please refer to the questions.

**Questions:**

* Is it worthwhile to increase the training cost of the extra in-context vector for the sake of reduced inference cost? I am curious about the trade-off between reduced inference time and additional training time.
* How much would performance drop if each layer shared the same trained bias vector?

**Limitations:**

Yes.

---

> ### Author Rebuttal · Authors · 2024-08-06
>
> **Q1: trade-off between reduced inference time and additional training time**
>
> We believe that performing additional lightweight training to reduce inference time is worthwhile.
> Firstly, the initial one-time training investment is minimal and yields substantial long-term benefits. For instance, in our local machine (3090 GPUs) tests on VQAv2 experiments, training L-ICV takes approximately **11 GPU hours**.  In comparison, inferring 20,000 data samples using 32-shot ICL takes about **13.4 GPU hours**, whereas L-ICV achieves the same inference in just **0.5 GPU hours**.  The time saved for 20,000 samples inference is comparable to the time spent on training. Consequently, the cumulative savings in inference costs can easily exceed this initial investment. With the increasing number of tokens required by scaled LMMs (e.g., IDEFICS v2), L-ICV's efficiency in saving memory and reducing inference costs becomes even more pronounced (Table E). This advantage makes L-ICV better suited for batch inference and enhances the throughput of inference services at scale.
> Secondly, unlike L-ICV, ICL's performance is highly unstable. While various methods exist to stabilize ICL performance—such as selecting ICDs similar to the query—these approaches inevitably incur additional computational and memory costs. In contrast, L-ICV offers stable performance while substantially reducing inference time
> In the upcoming revision, we will further discuss this issue, emphasizing the feasibility and necessity of lightweight training in exchange for faster inference.
>
> ---
>
> **Q2: How much would performance drop if each layer shared the same trained bias vector?**
>
> Thank you for your suggestion. We conducted experiments to investigate the impact of using a shared bias vector across all layers (as shown in Table D), and we observed a significant performance drop compared to ours: 20.6% on VQAv2 and 9.01% on OKVQA. This performance drop aligns with the findings in [22] that different Transformer layers play diverse roles in ICL (Lines 70-72). Also,  Equations (1-4) in Sec.3 show that L-ICVs are layer-specific.  Then using a single shared vector fails to capture the distinct information processed at each layer, leading to the observed decline in accuracy.
> Additionally, since the number of tuned parameters is limited to 32 × 4096 for layer-specific L-ICVs, using a shared 1 × 4096 ICV across layers does not substantially impact training speed. During inference, the computational cost of using a shared ICV versus non-shared ICV remains comparable.

---

### Official Review · Reviewer_VKDo · 2024-07-13

**Soundness:** 3
**Presentation:** 3
**Contribution:** 3
**Rating:** 7
**Confidence:** 3

**Summary:**

This paper introduces learnable in-context vector (L-ICV) that extends the standard ICV method by training it on task samples. The method achieves performance benefits similar to ICV and the training methodology enables the L-ICV to outperform while still preserving the benefits of few-shot in-context learning. This is achieved through the use of a KL-loss term, that preserves the behavior rather close to ICV, but allowing adaptation as necessary.

The evaluation of the method is done on OKVQA and VQAv2 datasets. For both of datasets the performance of L-ICV outperforms that of LoRA fine-tuning. Authors provide an explanation and namely that due to need to access external knowledge, preserving the in-context learning behavior is advantageous compared to performing LoRA fine-tuning that disrupts the behavior, particularly when using few samples.

**Strengths:**

I believe the authors present a well-grounded method extending the strengths of in-context learning through few-shot learning. I find the learning approach balances well the specific model behavior with the task needs.  This is achieved through

1) Strong theoretical grounding and explanation of the method, contrasting well the formulation and empirical results with other standard methods, such as ICV and LoRA.
2) In-depth visualizations, qualitative assessment of the differences between the methods, such as the t-SNE visualization, decoding of the top-10 tokens and performance graphs compared effect of varying numbers of samples on L-ICV and LoRA

**Weaknesses:**

There are a few weaknesses, when contrasting it in-depth with methods focused on ICL for visual-language models.

1) Ignoring some of the recent work on interpretation of in-context learning abilities (whether few-shot or zero-shot). It has been discussed as to whether the learning process performs task recognition (superficial format recognition, for example) or task learning (learning to map inputs to outputs) [[1]](https://arxiv.org/abs/2305.09731). The concept was extended to VLMs, by extending it to visual task recognition, studied particularly on VQA [[2]](https://arxiv.org/abs/2312.01571). It seems that VLMs suffer from image specific when doing ICL, such as short-cut inference (ignoring the image similarity, picking on similar question-answers) and others.

2) The performance of LoRA is intriguing. Although I understand the motivation to completely treat it as baseline and naively apply on the same amount of trainable parameters, I find the comparison forcefully made to fit the story of the paper. The choice of tasks that require external knowledge make it obvious that LoRA can't be a good fit. Here it would be fair to either add another task with knowledge contained or alternatively modify on what LoRA is applied (e.g. synthetic 32-shot ICL examples) to contrast on fair grounds with L-ICV.

[1] https://arxiv.org/abs/2305.09731
[2] https://arxiv.org/abs/2312.01571

**Questions:**

Questions directly map to weaknesses discussed at previous point.

1) Could authors ground the work in this line of interpretation of in-context learning and explain how L-ICV improves upon task recognition or task learning angles?

2) Could the authors extend their analysis to such a task (e.g. OcrVQA, DocVQA, ChartQA, or InfoVQA)? Alternatively, why is comparing naively LoRA with L-ICV tuned to the output behavior of few-shot in-context demonstrations a fair comparison at all?

I'm willing to raise my score if the authors help me better understand these angles.

**Limitations:**

Yes.

---

> ### Author Rebuttal · Authors · 2024-08-06
>
> **Q1 & W1: Task Recognition (TR) and Task Learning (TL)**
>
> Thanks for your recommendations of these studies. In our final discussion, we are happy to include these recent works on the interpretation of ICL abilities of LMMs. After thoroughly reviewing [A,B], we think that theoretically, L-ICV is more akin to TL. This is because during the training of L-ICV, we use randomly sampled 32-shot ICDs to encourage L-ICV to learn the common knowledge of the task (Lines 68-71), while overlooking the individual characteristics of the ICDs. In this way, L-ICV does not exploit the individual characteristics of the ICDs to narrow down the label space for predicting as TR mentioned in Sec.3.1 and Figure 1 of [B]. Moreover, Equations (1-4) in Sec.3 of our paper might provide a better mathematical explanation for TL. Specifically, in the latent space, the contextual information of ICDs  helps shift the query representation toward the direction of the correct answer. The role of L-ICV is to learn the most stable shift vector from various ICD combinations. In this way, L-ICV makes LMM less affected by short-cut inference since this often over-exploits the individual characteristics of the ICDs. The analyses in Line 302 "Hallucinations and Invalid Responses" show that L-ICV is less affected by yes/no hallucination, suggesting L-ICV is more robust that it does not simply copy yes/no from the ICDs as the answer.
> Additionally, we follow [B] to deploy the "new-mapping" experiment for testing TL ability, where the results are given in Table A. Specifically, when training L-ICV and testing, all "yes/no" labels are changed to "tiger/lion". When applying ICL, we also change "yes/no" to "tiger/lion". In this way, if L-ICV has good TL ability, it will learn the mapping between the correct answer to one new label instead of recognizing "yes/no". Results in Table A show that L-ICV achieves better results than ICL, suggesting it has good TL ability.
>
> [A] What In-Context Learning "Learns" In-Context: Disentangling Task Recognition and Task Learning
>
> [B] How to Configure Good In-Context Sequence for Visual Question Answering
>
> **Q2 & W2: Fair LoRA Comparisons**
>
> 1. First, we would like to clarify that in Section 4.3, the term "external knowledge" specifically refers to the OKVQA dataset, not to imply that "since both VQAv2 and OKVQA require external knowledge, LoRA has worse performance."  For VQAv2, which does not require external knowledge, LoRA performs reasonably well (as shown in Figure 4, main paper). However, our L-ICV still achieves superior results with a notable improvement of 2.36%.
> 2. We followed your suggestion to train LoRA in a fairer setting by using 32-shot ICDs and applying the same KL loss as our L-ICV. As shown in Table B, even with this modified design, LoRA's performance on VQAv2 and OKVQA remains significantly lower than that of L-ICV (49.48% vs. 58.54% in VQAv2 and 37.02% vs. 50.08% in OKVQA). Additionally, we evaluated LoRA with more parameters, as suggested by Reviewer fRTc, and the results are presented in Table B. Similarly, we find that increasing the parameter count does not bridge the performance gap between LoRA and our L-ICV, demonstrating the effectiveness of L-ICV's unique design.
>
>
> **W2: More Extension Tasks**
>
> We followed your advice to evaluate L-ICV on OcrVQA, which also does not require external knowledge. The results are shown in Table C, and we find that L-ICV outperforms LoRA and 32-shot ICL by 1.6% and 1.2% respectively. Additionally, we evaluated our performance on Image Captioning (Table C) and another multimodal model architecture, IDEFICS2-8B (Table E). These consistent improvements over 32-shot ICL and LoRA demonstrate that our method is highly effective and efficient across various tasks and models.

---

### Author Rebuttal · Authors · 2024-08-06

We gratefully thank all the reviewers for their valuable and constructive feedback. We are pleased to see that the reviewers recognize our motivation: to reduce inference time costs while maintaining the performance of in-context learning (ICL) for LMMs. We are encouraged to see that they find our method novel, insightful and strongly theoretically grounded (Reviewer VKDo, 8z8x, and BDyQ), our visualizations and qualitative assessments comprehensive and in-depth (Reviewer VKDo, 8z8x), our experiments solid and the results strong and effective (Reviewer 8z8x).

We address the concerns and questions in detail below and have appended a PDF file with tables and figures. **Note that to distinguish from the Tables and Figures in the submitted manuscript, the Table and Figure numbers in the rebuttal are marked with A, B, C, etc.**

 Based on these comments, we have summarized the common questions and our responses as follows:

1. **Generalization Ability:** We conducted supplementary experiments on other vision-language tasks and LMM to demonstrate the general applicability of our L-ICV method (Table C and Table E, to Reviewers VKDo fRTc, BDyQ).
2. **More Ablation Studies of L-ICV:** We conducted additional comparisons of L-ICV over the layer-shared, randomized and more-shot-trained ablation to see the effectiveness of our design. (Table D Table F, and Table G, to Reviewer 8z8x, fRTc)
3. **Training Cost and Inference Efficiency:** We provided detailed explanations on how our method can use a slight increase in training costs to achieve significantly faster inference times and better ICL performance, highlighting the significance of this trade-off (Table E, to Reviewer 8z8x, BDyQ).
4. **Comprehensive Comparisons with LoRA:** We show additional explanations and comparison experiments with LoRA,  considering the task does not requiring external knowledge, the different design and parameter count of LoRA (Table B, to Reviewer VKDo, fRTc).

We also address other specific concerns in separate responses.

---

### Decision · Program_Chairs · 2024-09-25

**Decision:**

Accept (poster)

**Comment:**

The paper introduces Learnable In-Context Vector (L-ICV), an approach that enhances in-context learning for multimodal models, specifically in Visual Question Answering (VQA) tasks. Reviewers commend the method for its efficiency and the way it balances performance with reduced computational costs, achieving results comparable to or better than LoRA, particularly on VQAv2 and OKVQA datasets (reviewers 1, 2). The paper is noted for its clear and concise presentation, with strong theoretical support and informative visualizations that help to distinguish L-ICV from other techniques (reviewers 1, 3). However, some concerns were expressed regarding the limited range of evaluations, as the method was primarily tested on one model and two datasets, which may affect its generalizability to other tasks and models (reviewers 3, 4). Additionally, reviewers suggested that the paper could benefit from a broader comparison with other in-context learning methods and a more detailed exploration of the trade-offs between the additional training effort and the savings in inference time (reviewers 2, 4). Overall, the reviewers agree that the paper provides a valuable contribution to the field, justifying its acceptance despite the identified limitations.